# Comprehensive Physiologically Based Pharmacokinetic Model to Assess Drug–Drug Interactions of Phenytoin

**DOI:** 10.3390/pharmaceutics15102486

**Published:** 2023-10-18

**Authors:** Leyanis Rodriguez-Vera, Xuefen Yin, Mohammed Almoslem, Karolin Romahn, Brian Cicali, Viera Lukacova, Rodrigo Cristofoletti, Stephan Schmidt

**Affiliations:** 1Center for Pharmacometrics and System Pharmacology at Lake Nona (Orlando), Department of Pharmaceutics, College of Pharmacy, University of Florida, Orlando, FL 32827, USA; leyanisrv80@gmail.com (L.R.-V.); xuefenyin@ufl.edu (X.Y.); almoslem.mj@gmail.com (M.A.); karolinromahn@gmail.com (K.R.); bcicali@ufl.edu (B.C.); 2Simulations Plus, Lancaster, CA 93534, USA; viera.lukacova@simulations-plus.com

**Keywords:** physiologically based pharmacokinetic modelling (PBPK), drug–drug interactions (DDIs), phenytoin, cytochrome P450 2C9 (CYP2C9), cytochrome P450 2C19 (CYP2C19)

## Abstract

Regulatory agencies worldwide expect that clinical pharmacokinetic drug–drug interactions (DDIs) between an investigational new drug and other drugs should be conducted during drug development as part of an adequate assessment of the drug’s safety and efficacy. However, it is neither time nor cost efficient to test all possible DDI scenarios clinically. Phenytoin is classified by the Food and Drug Administration as a strong clinical index inducer of CYP3A4, and a moderate sensitive substrate of CYP2C9. A physiologically based pharmacokinetic (PBPK) platform model was developed using GastroPlus^®^ to assess DDIs with phenytoin acting as the victim (CYP2C9, CYP2C19) or perpetrator (CYP3A4). Pharmacokinetic data were obtained from 15 different studies in healthy subjects. The PBPK model of phenytoin explains the contribution of CYP2C9 and CYP2C19 to the formation of 5-(4′-hydroxyphenyl)-5-phenylhydantoin. Furthermore, it accurately recapitulated phenytoin exposure after single and multiple intravenous and oral doses/formulations ranging from 248 to 900 mg, the dose-dependent nonlinearity and the magnitude of the effect of food on phenytoin pharmacokinetics. Once developed and verified, the model was used to characterize and predict phenytoin DDIs with fluconazole, omeprazole and itraconazole, i.e., simulated/observed DDI AUC ratio ranging from 0.89 to 1.25. This study supports the utility of the PBPK approach in informing drug development.

## 1. Introduction

Drug–drug interactions (DDIs) are often observed when multiple medications are co-administered, potentially leading to drug-induced toxicity or inefficacy [1,2]. DDIs frequently involve enzymatic systems, where one drug (perpetrator) can alter the metabolism and thus the exposure of a second drug (victim). As a result, regulatory agencies expect studies examining DDI risks between an investigational new drug and other drugs to be conducted as part of an adequate assessment of the drug’s safety and efficacy. However, it is neither time nor cost efficient to study all possible DDIs in head-to-head clinical trials [3]. Physiologically based pharmacokinetic (PBPK) simulation approaches have emerged as a valuable alternative for studying various DDI scenarios and are currently broadly accepted by industry and regulatory agencies around the globe [4]. Respective PBPK simulations are carried out using available in silico platforms. It is consequently important to have pre-built PBPK models for prototypical victim and perpetrator drugs, particularly for clinical index perpetrators and clinical index substrates, readily available to predict clinical DDIs of new drug products in order to streamline the drug development and regulatory evaluation process. When establishing these models, it is important to first evaluate the different pathways, separately, using prototypical victim and perpetrator drugs to, subsequently, better understand the relative contribution of each pathway to the overall metabolism of a drug.

Phenytoin, formerly diphenylhydantoin, is a first-generation anti-convulsant that is effective in the treatment of generalized tonic–clonic seizures, complex partial seizures and status epilepticus without significantly impairing neurological function. It is a Biopharmaceutics Classification System (BCS) class II drug [5]. Phenytoin is primarily metabolized via CYP2C9 (80–90%) and to a lesser extent via CYP2C19 (10–20%) to the inactive 5-(4′-hydroxyphenyl)-5-phenylhydantoin (p-HPPH) [6,7,8]. Given its extensive CYP2C9 metabolism, phenytoin can be considered an ideal probe drug for CYP2C9 [9]. CYP2C9 is a clinically important enzyme because it catalyzes approximately 20% of all phase-I reactions of currently prescribed small molecule drugs [10]. Additionally, phenytoin is considered as a strong clinical index inducer of CYP3A4 by the Food Drug Administration (FDA) [11], making it valuable for studying DDIs [12,13]. Patients undergoing chronic phenytoin treatment are at risk of DDIs when introducing medications primarily metabolized by CYP3A4 or medications that function as CYP2C9 inducers or inhibitors. Phenytoin can enhance the metabolism of co-administered CYP3A4 substrates, including estrogens [13], progestogens [13], voriconazole [14]], itraconazole [15], amiodarone [16], ritonavir [16], lopinavir [17], ivabradine [18], atorvastatin [19], nisoldipine [20], midazolam [21], quetiapine [22], digoxin [23] and cyclosporine [24]. Conversely, when combined with CYP2C9 and/or CYP2C19 inhibitors, such as fluconazole [25,26] and voriconazole [14], phenytoin blood levels will increase, elevating the risk of side effects. Lopinavir [17] and ritonavir [17], through CYP2C9 induction, can have the opposite effect, reducing phenytoin blood levels and potentially increasing the risk of seizures. Under the above considerations, the objective of this study was to develop a PBPK model for phenytoin to be used in the evaluation of DDIs either as an inducer of CYP3A4 or as a substrate of CYP2C9/CYP2C19.

There are several challenges to be overcome when attempting to establish and verify a PBPK model for phenytoin due to multiple sources contributing to the nonlinear pharmacokinetics of the drug. First, phenytoin is a poorly soluble drug, which results in dose-dependent oral bioavailability. Second, its reported unbound fraction covers a wide range (1–61%), affecting the distribution and clearance process, and ultimately unbound plasma concentrations [27,28,29,30,31,32,33]. Third, phenytoin’s clearance is subject to capacity-limited metabolism and autoinduction, resulting in dose-dependent nonlinearity in clearance [34]. Although several multiple-dose studies have been performed to evaluate phenytoin’s nonlinear pharmacokinetics, they (in part) suffer from study design limitations, making a clear distinction between the different sources of variability and nonlinearity difficult. In addition, phenytoin is a narrow therapeutic index drug. It is consequently important to appropriately characterize and predict its PK in order to select an optimal dosing regimen when given either alone or in combination with other drugs, particularly in light of the serious adverse event potential of the drug, such as decreased coordination, mental confusion, slurred speech and nervousness [5,34,35].

## 2. Materials and Methods

### 2.1. Software

GastroPlus^®^ version 9.8.2 (Simulation Plus, Lancaster, CA, USA) was used to develop and verify the PBPK model for phenytoin. The ADMET Predictor^®^ module was used to obtain in silico estimates of key physicochemical parameters from structures where experimentally determined values were not available or to provide an objective alternative to experimental data. The PBPKPlus^TM^ module was used to establish the systemic distribution and clearance of phenytoin. The metabolism module was used to account for the saturable metabolism of phenytoin. The advanced compartmental absorption and transit (ACAT™) model was used to simulate phenytoin in vivo in dissolution and absorption data for different oral formulations. The DDI Module was used to predict competitive inhibition, time-dependent inhibition and autoinduction drug–drug interactions with dynamic simulation. The data from the scientific literature were digitized using Graph Grabber version 2.0.2. The PK parameters after multiple doses were calculated using Phoenix^®^ WinNonlin^®^ version 6.4 (Certara USA Inc., Princeton, NJ, USA) and the goodness-of-fit plots were created with R 4.1.3 (the R foundation for Statistical Computing, Vienna, Austria 2021).

### 2.2. Clinical PK Data

Clinical plasma concentrations vs. time data after single and multiple dose administrations of phenytoin were collected and digitized from the literature. A total of 15 clinical datasets in healthy volunteers were used to develop and verify the phenytoin PBPK model. The data, summarized in Appendix A, were divided into a training dataset (*n* = 4) and a test dataset (*n* = 11). During model development, four studies were utilized to parameterize distribution, metabolism and absorption [8,36,37]. In addition, these studies allowed us to explore the nonlinear behavior and autoinduction mechanism of phenytoin. For model verification, 11 clinical studies of phenytoin after the administration of single and multiple doses ranging from 248 to 900 mg were used. Food effect on phenytoin PK and different oral formulation performances were also evaluated via simulation and comparison to counterpart clinical observations [15,25,26,38,39,40,41,42]. Following model verification, a total of 4 additional clinical studies were used for building the DDI aspects of the phenytoin PBPK model in the presence of fluconazole, omeprazole and itraconazole (Appendix A).

### 2.3. PBPK Model Development of Phenytoin

Figure 1 shows the workflow that summarizes overall analysis strategy.

The first step in developing the phenytoin PBPK model was a thorough review of the literature to collect all physicochemical parameters and pharmacokinetic/clinical information on phenytoin. All drug-dependent parameters were derived from in vitro or in vivo studies, with the exception of the diffusion coefficient, which was predicted with the ADMET Predictor^®^ module based on the molecular structure. Table 1 summarizes all parameter values of phenytoin.

Human organ weights, volumes and blood perfusion rates specific to subjects in each study (gender, mean age and mean body weight) were generated with the GastroPlus^®^ internal Population Estimates for Age-Related (PEAR) Physiology™ Module (Appendix A). Model development was started using the data after the intravenous administration of phenytoin. Drug tissue/plasma partition coefficients (Kps) were estimated using the Lukacova (Rodgers-single) method [52]. A local sensitivity analysis was conducted in GastroPlus^®^ to select the best fup and logP values due to the various values reported in the literature [27,29,30,43,44]. Exposure values after 250 mg i.v. infusion from Glazko et al., 1969 [36] were used as reference exposure to select the optimal parameter values (Appendix A). Systemic clearance value for the sensitivity analysis was set to 1.83 L/h [36], considering the similarity in all the CL reported after phenytoin i.v. administration in the literature [36,37,53]. These values were also used as the starting point to derive the unbound intrinsic clearance using a top-down approach [54]. Hepatic clearance from total systemic clearance of phenytoin included contributions from CYP2C9-mediated (major pathway) and CYP2C19-mediated (minor pathway) metabolism in the liver. Gut metabolism was considered negligible due to the low expression of these enzymes in the gut. The metabolic conversion with CYP2C9 and CYP2C19 was modeled using Michaelis–Menten kinetics with built-in expressions of CYP2C9 and CYP2C19 in the liver [55,56,57,58]. The contribution of each specific enzyme to the hepatic clearance was initially taken from the literature [6], and was also derived from Caraco et al., 2001 with the data from poor metabolizer subjects who carry two mutated CYP2C9 alleles allowing the optimization of the CYP2C19 Vmax [8]. Afterwards, Vmax in CYP2C9 was optimized based on a group of normal metabolizers from the same study [8]. Lastly, both optimized Vmax values were verified using the training IV data. The Km values in CYP2C9 and CYP2C19 were fixed using in vitro experiments in human liver microsomes as indicated in Table 1 [7]. The amount of phenytoin eliminated unchanged in urine was less than 5%. The renal clearance value was fixed at 0.015 L/h based on the literature [50].

In order to appropriately characterize the impact of formulation on phenytoin’s absorption and PK, particle size distribution values were fitted in GastroPlus^®^, using a parameter sensitivity analysis approach (PSA). The resulting particle size value falls within the reported range obtained from the two literature studies for phenytoin formulations [45,59]. The ACAT™ absorption model was established using the data from Gugler et al., 1976 [37] after a single 300 mg oral dose using the default parameters for passive transcellular absorption. The first-order model was used to describe phenytoin precipitation with a default value of 900 s for precipitation time. Considering the low and similar solubility of phenytoin free acid and phenytoin sodium between pHs 1–6, no significant precipitation during gastric transit was expected and the performance of the model was confirmed in simulations of a wide range of oral doses. The phenytoin dose-dependent nonlinear PK was investigated using a dose range of 200 to 900 mg in single and multiple i.v. and oral doses to elucidate the mechanism associated with nonlinearity. A power model was used along with the bioequivalence criteria proposed by Smith et al. (2000) to formally assess deviations from dose proportionality [60]. Finally, nonlinearity at multiple doses due to the autoinduction mechanism via CYP2C9-mediated metabolism was explored using values from the literature of Emax = 0.9 and EC50 = 15.3 µM [50] using the DDI module. The multiple-dose scenario used during model development corresponded to a dose of 300 mg of phenytoin administered once daily for 15 days [37]. The dose- and time-dependent autoinduction were then further investigated using simulations in the dose range of 200 to 900 mg. Above analyses supported the decision on whether the integration of autoinduction mechanisms is clinically significant and, thus, needed in the final model before the subsequent verification of the model with the external dataset.

### 2.4. PBPK Model External Verification

The external verification was completed graphically and numerically using the test dataset. The predicted plasma concentration–time profiles were compared to observed profiles and goodness-of-fit (GOF) plots were generated to compare the AUC from the time of drug administration to the time of the last concentration measurement (AUC_0-t_) and to infinity (AUC_0-inf_) and maximum plasma concentration (C_max_) values for all predicted versus observed values. Predictions within bioequivalent criteria and the 2-fold deviation from the observed values were used to assess the model’s performance. For a quantitative description of the model performance, the geometric mean fold error (GMFE) was calculated according to Equation (1). GMFE values < 2 were considered successful.
(1)$$GMFE=10^{\left(\sum \left|\log_{10}\frac{sim\;PK\;parameter}{obs\;PK\;parameter}\right|\right)/n}$$
where sim PK parameter represents simulated AUC or simulated C_max_ values, obs PK parameter represents observed AUC or observed C_max_ values and n represents the number of studies used for model verification.

### 2.5. Impact of Plasma Protein Binding on Phenytoin Exposure

The impact of plasma protein binding on phenytoin exposure was evaluated using the final model by simulating the plasma concentrations after 300 mg in three different fup scenarios, with both after single and multiple administration (15 days). Additionally, the percentage change of clearance and volume was calculated to evaluate the impact of differences in the unbound fraction of phenytoin PK, and then the possible need of optimizing this parameter in future applications of the model, mainly to see whether the unbound fraction is available in the subjects of this study.

### 2.6. DDI Simulations

To build the DDI aspects of the phenytoin PBPK model, phenytoin was treated as the victim in the presence of fluconazole and omeprazole and as the perpetrator drug in the presence of itraconazole, as outlined in Figure 2. The DDI module was used to predict competitive inhibition, time-dependent inhibition and induction mechanisms involved in the named DDIs using dynamic simulation. All drug-dependent parameters for the different drugs used here are provided in Appendix A.

DDI simulations involving fluconazole were conducted using the fluconazole model available in GastroPlus^®^ library. In the study by Touchette et al., 1992, a daily dose of 400 mg fluconazole was administered for 5 days; on day 4, a single dose of 250 mg phenytoin in a suspension formulation was co-administered [26]. In the study by Blum et al., 1991, a daily dose of 200 mg fluconazole was administered for 15 days. During days 10 to 12, a single dose of 200 mg phenytoin was co-administered orally; on day 13, 250 mg phenytoin was co-administered intravenously [25]. The fluconazole–phenytoin DDI was modeled as competitive inhibition of CYP2C9 and CYP2C19 metabolism. The unbound Ki values for fluconazole were 1.74 µM and 19.6 µM for CYP2C19 and CYP2C9, respectively [61].

DDI simulations with omeprazole were performed using the omeprazole model from GastroPlus^®^ library for a 300 mg of oral phenytoin on day 7 with omeprazole 40 mg during 9 days in CYP2C19 normal and intermediate metabolizers (NM and IM). The omeprazole–phenytoin DDI was modeled as mechanism-based inhibition and competitive inhibition of CYP2C19 was completed with omeprazole. K_I_, Kinact and IC50 values were 1.1 µM, 0.048 min^−^^1^ [62] and 8.4 µM [63], respectively.

DDI simulations with itraconazole were performed using the itraconazole model from GastroPlus^®^ library. Itraconazole 200 mg was administered on day 14 as single oral dose and phenytoin was given as 300 mg daily for 17 days. The induction of the CYP3A4-mediated itraconazole metabolism with phenytoin was described using a literature value for the Emax = 12.6 [50], whilst the EC50 = 3.7 µM was obtained from Fahmi et al., 2008 [51]. The EC50 value was confirmed with parameter sensitivity analysis.

### 2.7. DDIs Model Verification

The predictive performance of the DDI model was evaluated by comparing predicted to observed “victim drug” plasma concentration–time profiles, with and without the perpetrator drug. Additionally, predicted DDI AUC ratios (Equation (2) and DDI C_max_ ratios (Equation (3)) were calculated with the following equations (Equations (2) and (3)):(2)$$DI\;AUC\;ratio=\frac{AUC\;victim\;drug\;during\;coadministration}{AUC\;victim\;drug\;alone}$$
(3)$$DDI\;Cmax\;ratio=\frac{Cmax\;victim\;drug\;during\;coadministration}{Cmax\;victim\;drug\;alone}$$

We calculated the success criteria for maximum concentration ratio and AUC ratio predictions according to the criteria proposed by Guest et al. [64] (Equations (4)–(6)) as follows:Upper limit: Robs × limit (4)
Lower limit: Robs/limit (5)
Limit = (δ + 2(Robs − 1))/(Robs) (6)
where Robs represents the observed DDI ratio of C_max_ and AUC. If the observed ratios were less than 1, the reciprocal of the ratio was used for Robs. In this study, when δ = 1.25 and Robs = 1, this means that the limits on R are between 0.80 and 1.25. A coefficient of variation of phenytoin AUC and C_max_ of approximately 20% was used.

## 3. Results

### 3.1. PBPK Model of Phenytoin

The PBPK model of phenytoin showed accurate performance for both the training and test clinical datasets. These included single- and multiple-dose administrations ranging from 248 to 900 mg, under fasted and fed conditions, and different formulations. The comparisons of simulated to observed plasma concentration–time profiles of the training studies are shown in Figure 3.

Dose proportionality for the i.v. single dose could not be declared in the range of 200–900 mg because the higher bound of the confidence interval (1.1794) was outside the BE criteria (0.8516–1.1483). However, as the beta value (1.3264) was not so different from 1, dose linearity was re-evaluated in the case in the range of 200–600 mg, for which proportionality was declared. The nonlinear PK of phenytoin after single and multiple i.v and oral doses of 200 mg, 300 mg, 400 mg, 600 mg and 900 mg is depicted in Figure 4. Drug CL, after both single- (Figure 4a) and multiple-dose (Figure 4b) administration of phenytoin remains basically unchanged with just a minimal reduction (<10%) with increasing dose, except for the 900 mg i.v. administration (single and multiple dose). Therefore, the simulations showed a minimal contribution of the autoinduction mechanism on phenytoin exposure at clinical doses. Dose proportionality could not be declared for single nor multiple oral doses ranging from 200 to 900 mg when the BE limits were considered (Appendix A). The dose-dependent nonproportionality for oral doses was associated with an altered fraction dissolved and absorbed with a 35% and 41% reduction after the 900 mg dose in comparison with the 200 mg dose after single (Figure 4c) and multiple doses (Figure 4d), respectively.

The comparisons of simulated to observed plasma concentration–time profiles of the test studies are shown in Appendix A. The comparisons of predicted to observed AUC and C_max_ values of all studies including the GMFE are summarized in Appendix A. The GOF plot shown in Figure 5 concluded that 87% of the simulated AUC values and 100% of the simulated C_max_ values were within a 1.25-fold error of the respective observed values and all of them were within a 2-fold error of the respective observed values with the exception of the AUC from the clinical study by Touchette et al., 1992 [26]. Furthermore, the GMFE of C_max_, AUC_0-t_ and AUC_inf_ were 1.01, 0.91 and 0.91, respectively, showing an adequate prediction in 100% of the studies for C_max_, and 93.3% for both AUCs (14 out of 15 studies), confirming good model prediction.

### 3.2. Impact of Plasma Protein Binding of Phenytoin Exposure

Simulated plasma phenytoin concentrations over time with different plasma unbound fractions of 4.3%, 9.7% and 15% are shown in Appendix A. Exposures were slightly different when the unbound fractions were 4.3% and 15% in comparison with the 9.7% fraction considered in the final model. These slight differences in the plasma concentrations are in line with changes in Vd, as CL is almost constant, where there was a reduction of 42% for an fup of 4.3% and an increase of 41% for an fup of 15% in Vd, respectively, both after single and multiple oral doses.

### 3.3. DDI Simulations

Figure 6 and Appendix A illustrate the comparison of the predicted versus observed plasma concentration–time profiles from the DDI evaluations in the linear scale and the semilogarithmic scale, respectively. The DDI simulations recapitulated the impact of fluconazole, 200 and 400 mg, and omeprazole, 40 mg, on phenytoin PK (Figure 6a–d), and the impact of phenytoin as a CYP3A4 inducer on itraconazole PK (Figure 6e). The AUC and Cmax ratios obtained from the DDI dynamic simulations versus the respective ratios from the observations are presented in Appendix A.

Figure 7 depicts the GOF of the AUC and C_max_ ratios, where the simulated/observed DDI AUC ratio ranged from 0.89 to 1.25. Additionally, all the simulated/observed DDI C_max_ ratios were within the 1.25-fold deviation except for the itraconazole–phenytoin DDI model that fell outside the limits proposed by Guest et al. (C_max_ ratio = 2.14).

## 4. Discussion

In this study, we successfully developed and verified a whole body PBPK model of phenytoin. The PBPK model accurately predicted phenytoin exposure following administration of single and multiple doses ranging from 248 to 900 mg in fasted and fed scenarios and after different oral formulations. Furthermore, the model was applied for DDI simulations using different scenarios with fluconazole, omeprazole and itraconazole. This PBPK model adequately describes the metabolism of phenytoin using CYP2C9 and CYP2C19 with contributions of 73% and 27%, respectively, that are close to those reported in vitro studies (i.e., fmCYP2C9 = 80–90% and fmCYP2C19 = 10–20%) [6,7,8,9]. This model was developed for phenytoin to serve as a prototypical drug model to inform drug development, given that phenytoin is classified by FDA as a clinical-index inducer of CYP3A4 in DDI studies as well as a moderate sensitive substrate of CYP2C9 enzyme [11].

Several other mechanisms (not included in our PBPK model) have been described as being potentially involved in phenytoin excretion, such as the effect of the efflux transporter p-glycoprotein (P-gp), entero-hepatic recirculation, entero-enteric recirculation and intestinal excretion [65]. However, their clinical impact has not been demonstrated. Regarding the effect of P-gp, there is a controversy of its clinical relevance on phenytoin PK and the main effect seems to be at the blood–brain barrier level [66,67,68,69]. Additionally, phenytoin seems to only be a weak substrate of this transporter, which is in line with its high bioavailability (>70%) [70]. To our knowledge, all studies suggesting the involvement of entero-hepatic recycling were conducted using high doses (>1000 mg) of phenytoin, which seems to be confounded by the dose-dependent nonlinear kinetics of the drug. For example, the study from Mauro et al., 1987 aimed at studying the effect of multiple-dose activated charcoal on phenytoin elimination [71]. This study used doses of 15 mg/kg, which equates a total dose of 1200 mg for an 80 kg (mean weight of the study population) subject. The clearance in the absence of activated charcoal reported by Mauro et al. (0.9 L/h) was approximately two-fold lower than the clearance (~1.82 L/h) in other clinical studies at lower doses (~300 mg one daily or 150 mg twice daily) as reported Lim M. et al., 2004 [17] and Vlase L et al., 2012 [18]. In addition, the phenytoin dose recommended as a clinical dose in patients and DDI studies is 300 mg/day. It can be administered q.d., b.i.d or t.i.d., with no titration period needed [72]. Based on Haarst et al.’s clinical experience, they recommend 100 mg t.i.d. (total daily dose 300 mg) for at least 14 days. Considering all above information and the main purpose of the study of the PBPK model herein, the lack of inclusion of the above mechanism, the absorption and excretion mechanism will not impact the applicability of the current model for its use to predict metabolic DDI with phenytoin. The application of the current model to clinical scenarios, both with phenytoin as single agent and in DDI scenarios, was performed using a 9.7% value of the fup. However, high variability in plasma protein binding is reported in the literature, with values ranging from 1% to 61% [27,28,29,30,31,32]. The selection of the fup was completed based on a sensitivity analysis with the study from Glazko et al., 1969 [36]. It is important to highlight that the differences in the fup affect phenytoin exposure as shown in Appendix A, and the use of a fixed value could be a limitation, especially in subjects with altered plasma protein binding. However, our model was developed and verified using the data from healthy subjects in a dose range that includes the standard dose used in clinical DDI studies [72] with phenytoin, and a 9.7% value of the fup was appropriate for all the studies used in the internal and external verification of phenytoin alone and also in the DDI studies. Nevertheless, it is recommended to account for this factor and the possible need for fup optimization when the present PBPK model is intended to be used in future investigations, mainly in populations with altered plasma protein binding.

The PBPK model was applied for the prediction of phenytoin exposure from different formulations, which used either the sodium salt or the free acid form. A study by Serajuddin and Jarowski 1993 evaluating the potential differences in the pH-dependent solubility profile of phenytoin and its sodium salt showed that both have identical solubility profiles across the entire pH range [45]. Furthermore, the data from Dill et al., 1956 showed that phenytoin is a weak acid, which primarily exists in the acid form at pH values of ≤8 [39]. Upon oral administration, phenytoin consequently converts into its acid from and remains in this form throughout the GI tract. Therefore, even though the model did not explicitly differentiate between both, the salt and acid form, it sufficiently covers the behavior of phenytoin in the different pharmaceutical forms.

The particle size distribution parameters were determined by fitting them within GastroPlus^®^ using a PSA approach. The PSA results indicated that a mean particle radius range of 0.7 to 2.5 µm maintained consistent model performance. Dill et al., 1956 [39] provided information on the particle size range, reporting that most small particles had a diameter range of 1 to 3 µm, while a few larger particles averaged 7 × 26 µm. GastroPlus^®^ calculations based on the digitized data from Yasuji T. et al., 2005 [47] yielded a mean particle radius range of 0.748 to 1.594 µm for phenytoin mixed with polyvinylpyrrolidone and approximately 13.897 µm for the mean particle radius of the raw material. Given this information, a mean particle radius of 2.5 µm was chosen, with a fixed standard deviation of 0.75 to account for 30% of variability, and employing 4 bins. This decision was supported with the accurate predictions of phenytoin exposure for a variety of oral dosing scenarios (Figure 5) and accurate predictions of the food effect, with predicted versus observed ratios of PKfed/PKfasted for AUC0-inf, AUC0-tau and Cmax at 0.92, 0.89 and 0.85, respectively.

Our PBPK model accurately captured the plasma concentration–time profiles of phenytoin from different studies with AUC and C_max_ ratios within the 1.25-fold deviation with the exception of the study from Fraser et al., 1980 (Appendix A). The PK profiles in this study were measured in six healthy male volunteers after receiving a single 900 mg oral dose administered as nine capsules of phenytoin of 100 mg. Phenytoin was administered after an 8 h fasting period and patients received food 3 h after drug administration. C_max_ was reached after approximately 6 h and was maintained for about 10 h, indicating a saturable absorption, most likely due to poor aqueous solubility, and resembling the administration of an extended-release product. Our model predicts a decrease of the fraction absorbed by 35% and 41% after a single and multiple 900 mg oral dose in comparison to 200 mg. Simulations in the fed state after 3 h underpredicted phenytoin plasma concentrations; however, model predictions were still within the 2-fold error range [40]. Additionally, it is important to highlight the increased variability that may come from the administration of nine different capsules which was not accounted for by the model. It should further be noted that the AUC ratio of the study by Touchette et al., 1992 (Appendix A) was outside the 2-fold error range. This study tested a single 250 mg dose of phenytoin suspensions in nine healthy volunteers. Given that phenytoin is a BCS class II drug (i.e., low soluble and highly permeable drug), the absorption might be enhanced in the suspension formulation [26]. However, the Cmax was within the 1.25-fold error range of the observed data.

Simulations performed to evaluate the drug’s nonlinear pharmacokinetics showed a 35% and 41% decrease in the fraction absorbed after single and multiple 900 mg administration of phenytoin with respect to the lower evaluated dose of 200 mg, which may be explained by the poor aqueous solubility of phenytoin. The decrease of the fraction absorbed by phenytoin with increasing dose is related to a decrease in the fraction dissolved. Phenytoin has a dissolution rate-limited absorption because of its low solubility. The solubility of phenytoin is 0.04 mg/mL @ pH 3.29, which is less than the calculated phenytoin concentration in a glass of water at the lowest dose (i.e., 200 mg/250 mL = 0.8 mg/mL). As the contribution of gut metabolism was considered negligible, the Fa% curve overlaped the FD% curve (Figure 4c,d). On the other hand, the CYP2C9 enzyme in the liver is most likely to remain unsaturated when phenytoin reaches systemic circulation because the maximum plasma phenytoin concentration of 12 µM is smaller than the Km values of this major CYP2C9 enzyme (Km _CYP2C9_ is 14.6 µM). In addition, the F% curve resembles the FD% curve. Furthermore, after both single- and multiple-dose administration, simulations did not show a relevant change in drug CL, even when autoinduction mechanisms were integrated in the model. Although some of the literature highlights that phenytoin induces its own metabolism using CYP2C9 and CYP2C19 after multiple-dose administrations [34], the clinical effect of phenytoin autoinduction has been debatable [73,74]. Our PBPK model supports the latter, with accurate simulation of both single- and multiple-dose studies using the model without autoinduction. Under the above considerations, we concluded that the nonlinear PK of phenytoin at a single dose is related with a reduced dissolution of phenytoin at higher doses and thus a reduced fraction absorbed. This conclusion is in line with Fagiolino and Ibarra, 2021 who highlighted the mechanism involved in the dose- and time-dependent nonlinear pharmacokinetics of phenytoin, which concluded that incomplete dissolution in the digestive tract was responsible for the lack of dose proportionality between 400 to 1600 mg [75].

The PBPK model was further evaluated via the prediction of fluconazole–phenytoin DDI studies. Fluconazole is a moderate inhibitor of CYP2C9 and a strong inhibitor of CYP2C19 (with Ki values of 19.6 µM and 1.74 µM, respectively) in humans (Kunze et al., 1996) [61,76] after calculating the unbound fraction (0.87) [76]. The Ki value of CYP2C19 was measured in an in vitro experiment while the Ki value of CYP2C9 was measured in an in vivo human trial. Using these updated values, our PBPK model was able to optimally predict the impact of fluconazole on the metabolism of phenytoin. The simulated/observed DDI AUC_0-t_ ratios and DDI Cmax ratios were all within the 1.25-fold range for both studies, i.e., Blum et al., 1991 and Touchette et al., 1992 [25,26].

The contribution of CYP2C19 to phenytoin metabolism was further confirmed by the prediction of DDIs between omeprazole and phenytoin. Omeprazole is identified as a weak inhibitor of CYP2C19 by the FDA classification [11]. According to Shirisaka et al., 2013 [63], omeprazole inhibits phenytoin metabolism through two mechanisms simultaneously, competitive inhibition (IC50_-rev-*invitro*,total_ = 8.4 µM) and mechanism-based inactivation (Ki_irr-*invitro*,unbound_ = 1.1 µM and K_inact_ = 0.048 min^−1^). The impact of omeprazole on phenytoin by our model is well described by using both the omeprazole normal metabolizer PBPK model (NM) and the intermediate metabolizer model (IM), with simulated/observed DDI AUC ratios of 1.12 and 1.21, respectively, and simulated/observed DDI Cmax ratios of 1.02 and 1.04, respectively. The accurate predictions indicated that the contribution of the CYP2C9-mediated metabolic pathway is well described in our PBPK model.

The phenytoin induction effect on CYP3A4 was evaluated via the itraconazole–phenytoin DDI study where a single dose of itraconazole was co-administered after multiple doses of phenytoin. Phenytoin is classified as a strong inducer of the CYP3A4 enzyme [11], capable of significantly reducing the systemic exposure of other drugs metabolized with CYP3A4. This may cause sub-therapeutic levels, especially for drugs with a narrow therapeutic index. Therefore, it is meaningful to establish a DDI model to evaluate the phenytoin induction effect on the CYP3A4 enzyme. The induction parameters of phenytoin, Emax EC50 and Emax/EC50 on CYP3A4 from the literature are largely variable (1.9–29.04, 3.7–147 and 0.1–3.41, respectively) [51,77,78,79]. We fixed the Emax value at 12.6, based on the literature [50], and ran a parameter sensitivity analysis to find the optimal EC50 value to capture the itraconazole plasma concentration profile. An EC50 = 3.7 µM provided a good model performance with a simulated/observed DDI AUC ratio of 0.89 and a simulated/observed DDI Cmax ratio of 2.14. However, it is important to highlight that itraconazole exhibits large inter-individual variability, with clinical observed ratios of Cmax ranging from 6.7% to 43.8%, which covers our predicted value of 36.2%. This discrepancy is most likely caused by the dose-dependent behavior of itraconazole [80].

The phenytoin PBPK model can be applied to investigate and predict DDI scenarios with phenytoin as a CYP3A4 inducer and as a CYP2C9 and CYP2C19 sensitive substrate as well as to support dose recommendation for untested DDI clinical scenarios. As phenytoin is prescribed for the long-term treatment of epilepsy, chances are high for it to be co-administered with other interacting drugs. In this case, the in silico evaluation of DDIs’ clinical scenarios could help to identify safety and efficacy concerns in patients undergoing phenytoin therapy.

## 5. Conclusions

A PBPK model was developed in GastroPlus^®^ for the evaluation of a DDI with phenytoin, which included the contribution of CYP2C9 (73%) and CYP2C19 (23%) to its metabolism. The model reliably described the phenytoin single- and multiple- dose PK for the explored scenarios with all simulated/observed Cmax and AUC0-t ratios within a two-fold deviation. The presented PBPK model was able to successfully reproduce the DDIs with fluconazole, omeprazole and itraconazole. The model can be considered to be a verified substrate of CYP2C9 and CYP2C19, and an inducer of CYP3A4 for its use in DDI prediction. The integration of this model in the software library will allow drug developers to understand DDI mechanisms, design clinical trials and even support drug label information in untested clinical scenarios. This study supports the utility of the PBPK approach in informing drug development.

## Figures and Tables

**Figure 1 pharmaceutics-15-02486-f001:**
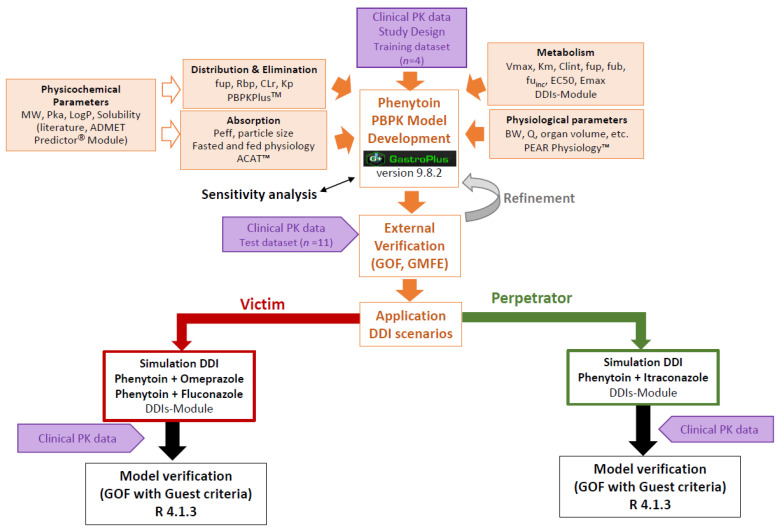
Workflow of the phenytoin PBPK modeling strategy: “model development and verification” on the top of the graph, and “model application to DDI scenarios” on the bottom of the graph. PBPK: physiologically based pharmacokinetics; MW: molecular weight; logP: partition coefficient; pKa: acid dissociation constant; Kp: tissue plasma partition coefficients; fup: fraction unbound in plasma; fub: fraction unbound in blood; fu_inc_: the free fraction of the compound in the microsomal incubation; Rbp: Blood: plasma concentration ratio; CYP: cytochrome P450; Km: Michaelis–Menten constant; Vmax: maximum reaction velocity; Emax: maximum effect; EC50: half-maximal effective concentration; Peff: effective permeability; GMFE: geometric mean fold error; GOF: goodness-of-fit plot.

**Figure 2 pharmaceutics-15-02486-f002:**
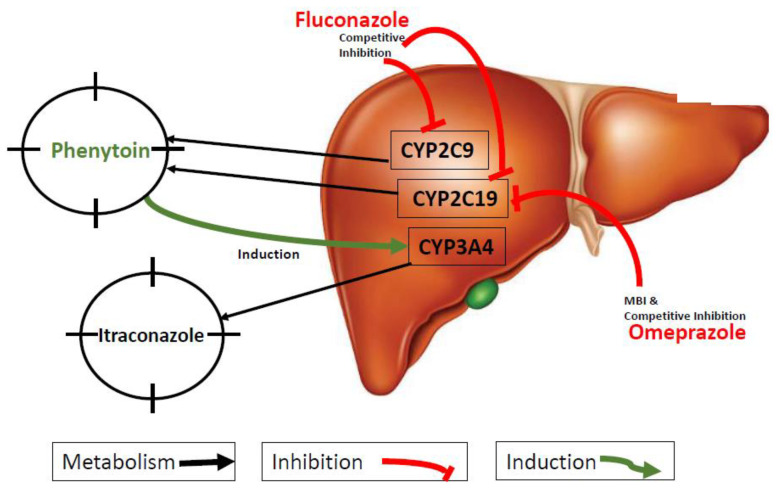
Overview of the modeled DDIs for phenytoin. Black arrow means sensitive victim in the specific metabolic pathway for phenytoin (CYP2C9 and CYP2C19) and itraconazole (CYP3A4). Red curve line means inhibition, e.g., fluconazole is a competitive inhibitor of phenytoin’s CYP2C9- and CYP2C19-mediated metabolism. Omeprazole inhibits the CYP2C19-mediated enzymatic conversion of phenytoin via competitive inhibition and mechanism-based inactivation (MBI). Green curve means the induction, e.g., phenytoin induces the CYP3A4-mediated enzymatic biotransformation of itraconazole.

**Figure 3 pharmaceutics-15-02486-f003:**
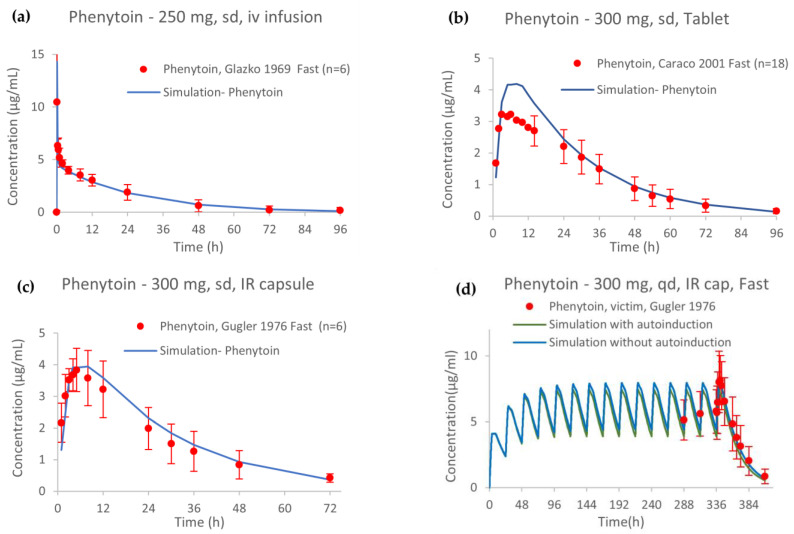
Model simulations of phenytoin concentration–time profiles from four studies in the training dataset (**a**–**c**) after a single dose (250 and 300 mg), for establishing distribution, metabolism and absorption phases, respectively. (**d**) Simulated profiles of phenytoin in comparison to the observed data after administering 300 mg in multiple doses for exploring phenytoin autoinduction mechanism [8,36,37]. Observed data are shown as red dots ± SD, and simulations are shown as blue or green solid lines. sd: single dose; iv: intravenous; IR: immediate release; qd: once daily.

**Figure 4 pharmaceutics-15-02486-f004:**
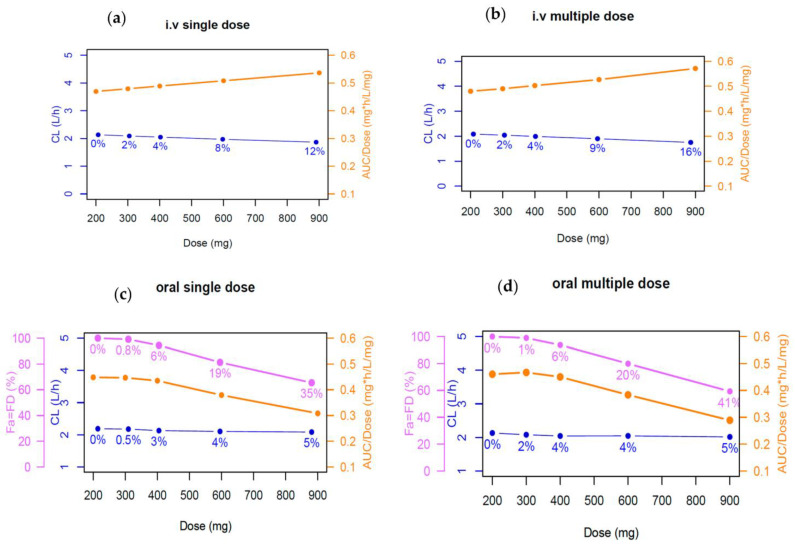
Nonlinear PK exploration of phenytoin after single and multiple i.v and oral doses of 200, 300, 400, 600 and 900 mg from Gugler et al., 1975 [37]. (**a**) i.v single dose, (**b**) i.v multiple dose, (**c**) oral single dose and (**d**) oral multiple dose. Orange line represents the area under curve/dose (AUC/Dose) overdose, blue line represents the clearance overdose, and purple line depicts the fraction dissolved (FD) and fraction absorbed (Fa) overdose. The percentage values describe the reduction changes of the absorption and elimination from dose of 200 mg.

**Figure 5 pharmaceutics-15-02486-f005:**
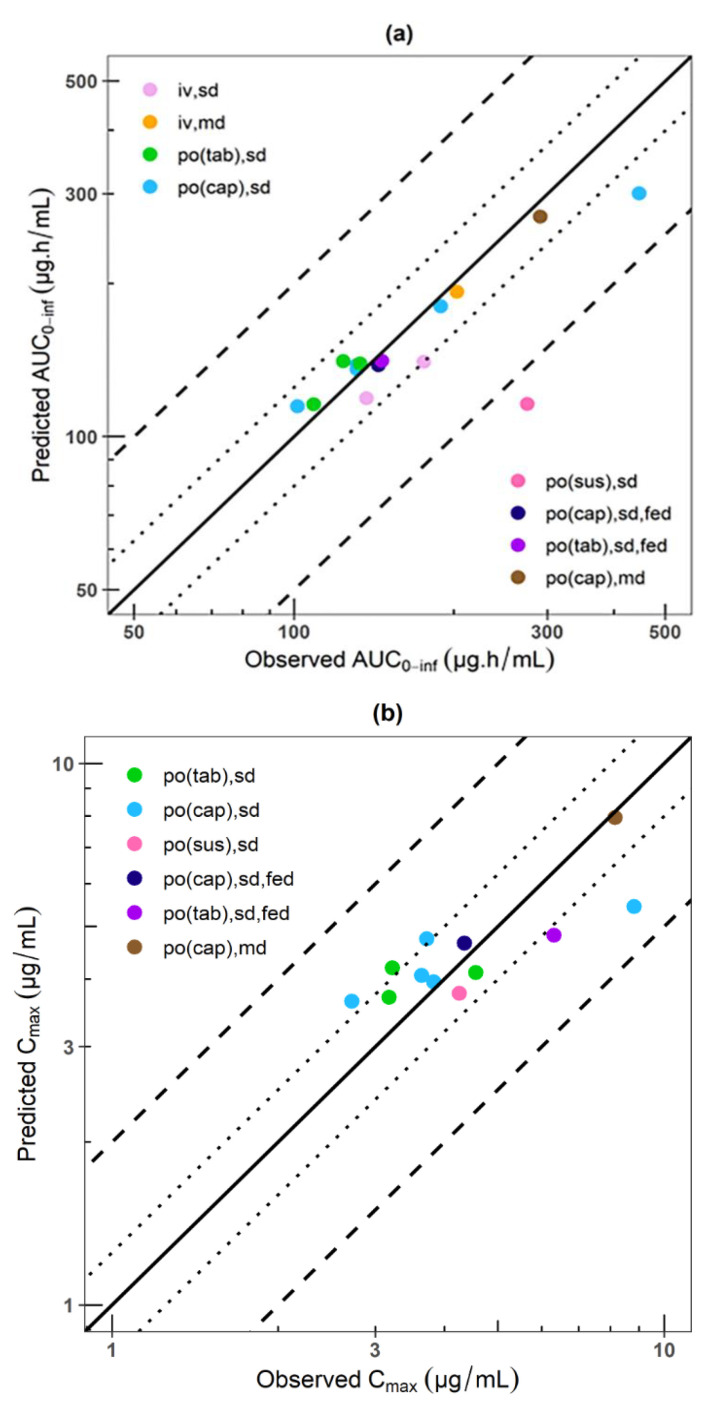
Phenytoin PBPK model. Comparison of predicted to observed (**a**) AUC0-inf values and (**b**) Cmax values of all analyzed studies. The line of identity is shown as a solid line; 1.25-fold deviation is shown as a dotted line; 2-fold deviation is shown as a dashed lines. Cmax: maximum concentration; AUC: area under the curve; IV: intravenous; MD: multiple dose; PO: oral administration; SD: single dose.

**Figure 6 pharmaceutics-15-02486-f006:**
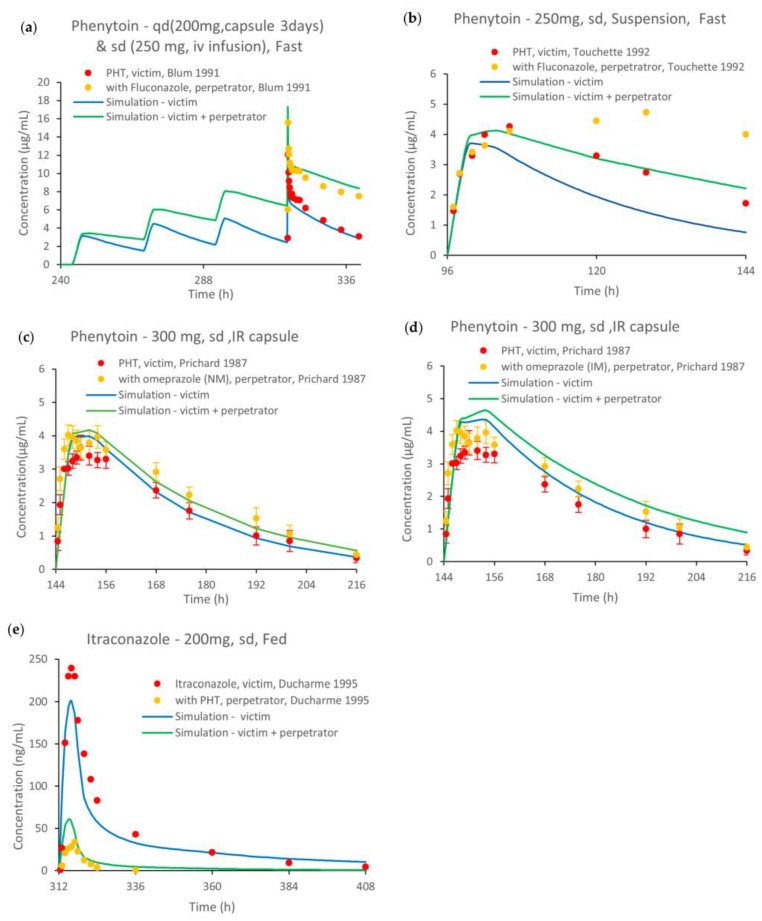
Simulated and observed phenytoin DDIs. Concentration–time profiles of phenytoin in comparison to observed data after a (**a**) multiple and (**b**) single dose with and without fluconazole (200 and 400 mg), respectively, (**c**) phenytoin with and without omeprazole 40 mg in normal CYP2C19 metabolizers (NM) and (**d**) phenytoin with and without omeprazole 40 mg in intermediate CYP2C19 metabolizers (IM). (**e**) itraconazole concentration–times profiles before and during phenytoin co-administration [15,25,26,28]. Observed data are shown as red and yellow dots, and simulations are shown as blue and green solid lines. q.d.: once daily; sd: single dose; PHT: phenytoin; IR: immediate release.

**Figure 7 pharmaceutics-15-02486-f007:**
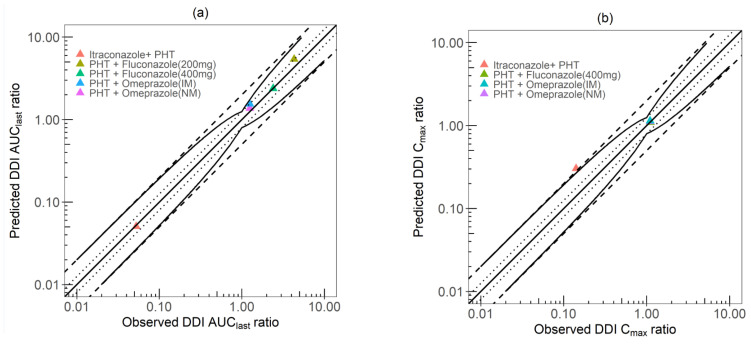
DDI PBPK model performance of phenytoin. Simulated vs. observed DDI AUC ratio is shown in (**a**) and simulated vs. observed DDI Cmax ratio is shown in (**b**). The line of identity is shown as a straight solid line; 1.25-fold deviation is shown as a dotted line; 2-fold deviation is shown as a dashed line. The curve solid lines show the prediction success limits proposed by Guest et al. allowing for 1.25-fold variability of the DDI ratios [64]. AUC: area under the curve; Cmax: maximum concentration.

**Table 1 pharmaceutics-15-02486-t001:** Key physicochemical and biopharmaceutical parameters of phenytoin PBPK model.

Parameter	Value	Reference
Molecular weight (g/mol) logP	252.27 2.21	Stella VJ et al., 1998 [43] Poulin et al., 2000 [44]
Diffusion coefficient (cm^2^/s)	0.86 × 10^−5^	ADMET Predictor v.10.0
pKa	8.249	Estimated from Solubility vs. pH profile ^a^
Reference solubility (mg/mL) at pH 3.29	0.04	Serajuddin et al., 1993 [45] and Chiang et al., 2013 ^b^ [46]
Particle radius (μm)	2.5 (SD = 0.75, Bins = 4)	Dill et al., 1956 [39] and Yasuji et al., 2006 [47]
Drug particle density (g/mL)	1.2	GastroPlus default value
Mean precipitation time (s)	900	GastroPlus default value
CaCo2 apparent permeability (P_app_), cm/s	34.3 × 10^−6^	Pade et al., 1998 [48]
Solubility (mg/mL, SGF at pH 4 at 0 mM) Solubility (mg/mL, FaSSIF at pH 6.4 at 10 mM) Solubility (mg/mL, FeSSIF at pH 6.4 at 20 mM) Solubility (mg/mL, FeSSIF at pH 6.4 at 30 mM) Solubility (mg/mL, FeSSIF at pH 6.4 at 40 mM)	0.04 0.0546 0.0764 0.1183 0.1392	Stella VJ et al., 1998 [43]
**Distribution**		
Kp calculation method		Lukacova (Rodgers-single)
fut calculation method		fut = S + 9.5v. (default)
Tissues		Perfusion limited
Blood: plasma concentration ratio (R_bp_)	1.33	Kong et al., 2014 [49]
Plasma protein binding (fup), %	9.7	Fitted from Peterson et al., 1982 [30]
Renal Clearance (CLfilt), L/h	0.015	Almond et al., 2016 [50]
**Metabolism (in vitro values converted to in vivo—Enzyme table)**		
CYP 2C19 Km, mg/L	5.474	Giancarlo et al., 2001 [7]
CYP 2C19 Vmax, ×10^−4^ mg/s/mg-enzyme	2.0042	Fitted to in vivo data from Caraco et al., 2001 [8]
CYP 2C9 Km, mg/L	3.316	Giancarlo et al., 2001 [7]
CYP 2C9 Vmax, ×10^−5^ mg/s/mg-enzyme	6.4531	Fitted to in vivo data from Caraco et al., 2001 [8]
**Induction**		
Emax (CYP3A4)	12.6	Almond et al., 2016 [50]
EC50*_invitro_*_,T_ (CYP3A4), µM	3.7	Fahmi et al., 2008 [51]
Emax (CYP2C9)	0.9	Almond et al., 2016 [50]
EC50*_invitro_*_,T_ (CYP2C9), µM	15.3	Almond et al., 2016 [50]
fu*_invitro_*, %	89.9	Calculated with Hallifax–HLM method

^a^: pKa is fitted to the combined pH solubility data from Chiang P et al., 2013 [46] and Serajudding et al., 1993 [45]. ^b^: the data point was selected from the combined pH solubility data used to calculate pKa. Appendix A shows the full solubility vs. pH profile from Serajuddin et al., 1993 and Chiang P et al., 2013 [45,46]. Emax is the maximum induction effect, EC50 is the concentration for half maximal induction. logP: octanol/water partition coefficient; pKa: acid dissociation constant; FaSSIF: fasted state simulated intestinal fluid; FeSSIF: fed state simulated intestinal fluid; Kp: tissue plasma partition coefficient; fut: fraction unbound in tissue; CYP: cytochrome P450; Km: Michaelis–Menten constant; Vmax: maximum reaction velocity.

## Data Availability

Not applicable.

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
