# Peer review of "Comprehensive Physiologically Based Pharmacokinetic Model to Assess Drug–Drug Interactions of Phenytoin"

_pharmaceutics, 2023, doi:10.3390/pharmaceutics15102486_

Round 1
Reviewer 1 Report
The manuscript is organized and written very well. Easy to read. The PBPK model development process is clearly described. The reviewer appreciated the tremendous work.
The reviewer has the following comments that the authors should address in their model before considering its acceptable for publication:
(1) The current model was developed for phenytoin free base, but mostly using data from phenytoin sodium. It is not clear to the reviewer how the authors have specified the differences between the free base and the salt form during the model development. For all the observed data collected from literature, it is essential to separate the PK data of free base from those of salt form.
(2) The current model is lacking mechanisms for absorption, though the model is developed for the purpose of DDI assessment. Phenytoin is a drug of low solubility and high potential of precipitation. The only information about phenytoin precipitation mentioned in the manuscript is that the GastroPlus default mean precipitation time (900s) was used. The current model needs to incorporate the mechanistic precipitation model contributing to its dose nonlinearity.
(3) The current model used the top-down approach to assign the total systemic clearance after IV administration to the hepatic metabolism and the renal excretion. This is a big assumption, and it is not correct from the viewpoint of the reviewer, because phenytoin is a P-gp substrate (although weak), of high passive permeability and long T1/2, therefore it has the potential of entero-hepatic recirculation, entero-enteric recirculation and intestinal excretion after IV administration (activated charcoal can significantly decrease phenytoin T1/2, https://pubmed.ncbi.nlm.nih.gov/3662159/). The systemic clearance after IV administration should include these components.
Below are minor comments:
· Line 22: “248 to 900mg”, the reviewer did not find any plot of simulation with 248mg, except some 250mg. The lowest dose 200mg was mentioned in the manuscript. Please clarify why 248 was specified here and in the main text.
· Line 44: “pre-build”, pre-built
· Line 86: “in vivo”, should this be in vitro? Phenytoin in vivo dissolution and absorption data are not ready to incorporate into ACAT.
· Figure 1, “B:P” should be Rbp.
· Table 1, EC50invitro,T(CYP3A4)=3.7uM, Fahmi 2008 was cited, but based on the description in line 246, EC50=3.7uM was fitted value, not from Fahmi's paper.
· Table 1, EC50invitro,T(CYP2C9)=15.3uM, Almond 2016 was cited. Please confirm if it is right reference here.
· Line 139: “B:P” should be Rbp.
· Line 140: blood. What and how blood parameters were used in Kp prediction in Lukacova-method?
· Figure 2: Omeprazole is also metabolized by 3A4. Has the model considered the impact of phenytoin-induced 3A4 induction on the PK of omeprazole?
· Figure 6: For all DDI simulations, please provide semi-log plots as well, such that the elimination phase and the impact due to CYP-mediated DDI can be well demonstrated.
· Figure 6(b): any comments on the model performance with respect to suspension formulation?
Author Response
-
Reviewer-1:
- The current model was developed for phenytoin free base, but mostly using data from phenytoin sodium. It is not clear to the reviewer how the authors have specified the differences between the free base and the salt form during the model development. For all the observed data collected from literature, it is essential to separate the PK data of free base from those of salt form.
Response: We thank reviewer for his/her insightful comment. Phenytoin is a Biopharmaceutics Classification System class II drug with low solubility and high permeability, i.e., it exhibits dissolution rate-limited absorption. A study from Serajuddin and Jarowski 1993 evaluating potential differences in the pH-dependent solubility profile of phenytoin and its sodium salt showed that both have identical solubility profiles across the entire pH range as depicted in Figure 1 below. While we conceptually agree with the reviewer’s comment, we consequently don’t think that this is of practical concern for our study.
Furthermore, data from Dill et al. 1956 showed that phenytoin is a weak acid, which primarily exists in the acid form at pH values ≤ 8. Upon oral administration, phenytoin consequently converts into its acid from and remains in this form throughout the GI tract. Therefore, we are of the believe that our PBPK model sufficiently covers the behavior of phenytoin and its sodium salt. We added respective language to the manuscript to make this point clearer. (See line 489)
- The current model is lacking mechanisms for absorption, though the model is developed for the purpose of DDI assessment. Phenytoin is a drug of low solubility and high potential of precipitation. The only information about phenytoin precipitation mentioned in the manuscript is that the GastroPlus default mean precipitation time (900s) was used. The current model needs to incorporate the mechanistic precipitation model contributing to its dose nonlinearity.
Response There seems to be some confusion here because this aspect is considered in the current model. The PBPK model captures well phenytoin PK for oral doses ranging from 248 to 900 mg without any obvious bias in the prediction errors with dose. That confirms that the simpler precipitation model (first order) is sufficient and the available did not contain enough information to parameterize/fit mechanistic nucleation model.Considering the low and similar solubility of phenytoin and phenytoin sodium between pHs 1 - 6 (see Figure 1 above), we do not expect significant precipitation during gastric transit. Instead, we expect slow dissolution due to the low solubility. We revised the manuscript accordingly to make this aspect clearer. (See line 206)
- The current model used the top-down approach to assign the total systemic clearance after IV administration to the hepatic metabolism and the renal excretion. This is a big assumption, and it is not correct from the viewpoint of the reviewer, because phenytoin is a P-gp substrate (although weak), of high passive permeability and long T1/2, therefore it has the potential of entero-hepatic recirculation, entero-enteric recirculation and intestinal excretion after IV administration (activated charcoal can significantly decrease phenytoin T1/2, https://pubmed.ncbi.nlm.nih.gov/3662159/). The systemic clearance after IV administration should include these components.
Response: Thank you for this insightful comment. While we conceptually agree with this comment, we believe that it is of limited clinical relevance because our PBPK model has been developed and verified in the context of use for DDI studies. We developed the model using in vitro enzyme kinetic information for each enzyme contributing to hepatic clearance from the literature [7] and also calibrated Vmax for CYP2C19 using poor metabolizer data from Caraco et al. 2001 [9]. The model was then extensively verified with data from multiple studies from the literature to evaluate its predictive performance under various conditions. We also evaluated the impact of autoinduction as there is conflicting data in the literature but decided to not include it into the model because clinical multiple dose data in healthy volunteers (i.e., target population for DDI studies) did not support autoinduction.
To our knowledge, all studies suggesting the involvement of entero-hepatic recycling were conducted using high doses (>1000 mg) of phenytoin, which seems to be confounded by the dose-dependent non-linear kinetics of the drug. For example, the study from Mauro et al. 1987 aimed at studying the effect of multiple-dose activated charcoal on phenytoin elimination. This study used doses of 15mg/kg, which equates a total dose of 1,200 mg for an 80kg (mean weight of the study population) subject. The clearance in the absence of activated charcoal reported by Mauro et al. (0.9 L/h) was approximately 2-fold lower than the clearance (~1.82 L/h) in other clinical studies at lower doses (~300 mg one daily or 150mg twice daily) as reported Lim M. et al 2004, and Vlase L et al 2012. In addition, Phenytoin dose recommended for DDI studies is 300mg/day. It can be administered q.d., b.i.d, or t.i.d., with no titration period needed. (Haarst A. et al 2023). Based on Haarst et al clinical experience, they recommend 100 mg t.i.d. (total daily dose 300 mg) for at least 14 days “. Given the context of use, we are consequently confident that our developed model can appropriately characterize DDI scenarios.
Regarding the effect of p-glycoprotein (P-gp), there is a controversy on the clinical relevance of P-gp on phenytoin PK and the main effect seems to be at the BBB level. Considering the objective of the present study, the lack of in vitro information on the transporter kinetics for phenytoin, and the consideration that phenytoin seems to be a weak substrate of P-gp . Furthermore, the reported oral bioavailability for phenytoin ranges between 70-100%, which suggests that the impact of P-gp on phenytoin’s PK is negligible and irrelevant in the context of use. We revised the manuscript accordingly to make this aspect clearer. (See line 452)
Below are minor comments:
- Line 22: “248 to 900mg”, the reviewer did not find any plot of simulation with 248mg, except some 250mg. The lowest dose 200mg was mentioned in the manuscript. Please clarify why 248 was specified here and in the main text.
Response: This plot is the Figure S5 b of Supplementary Materials. This Clinical study is from Smith TC et al 1976. they studied absorption and metabolism of phenytoin from tablets and capsules using dose of 250 and 248 mg, respectively.
Smith, T.C.; Kinkel, A. Absorption and Metabolism of Phenytoin from Tablets and Capsules. Clin. Pharmacol. Ther. 1976, 20, 738–742, doi:10.1002/cpt1976206738.
Line 44: “pre-build”, pre-built
Response: We thank reviewer for pointing this out. We updated the manuscript accordingly.
- Line 86: “in vivo”, should this be in vitro? Phenytoin in vivo dissolution and absorption data are not ready to incorporate into ACAT.
Response: We thank reviewer for pointing this out. In this case, we changed the word "incorporate" to "simulate" because the ACAT model simulates/predicts in vivo processes. We had properties that when linked to physiological intestinal model (ACAT) predict in vivo dissolution and absorption. We updated the manuscript accordingly.
- Figure 1, “B:P” should be Rbp.
Response: We thank reviewer for pointing this out. We updated the manuscript accordingly.
- Table 1, EC50invitro, T(CYP3A4)=3.7uM, Fahmi 2008 was cited, but based on the description in line 246, EC50=3.7uM was fitted value, not from Fahmi's paper.
- Response: The value of EC50=3.7uM is from Fahmi’s paper, please you can see the screenshot of the table I from Fahmi et al 2008.
In addition, we conducted a sensitivity analysis considering different in vitro values for phenytoin-mediated induction of CYP3A4, ranging from 3.6 to 51.3 µM. These values were obtained from Almond et al. 2016, Amanda et al. 2016, Fahmi el al. 2008, Nagai et al. 2018, Faucette et al. 2004, Zhan 2014 and McGinnity et al. 2009.
Table 1, EC50invitro, T(CYP2C9)=15.3uM, Almond 2016 was cited. Please confirm if it is right reference here.
- Response: We confirm this value comes from Almond 2016 in the Supplementary Material section, Table 1. In addition, Ban Ke A et al. 2013 reported the same value in Table 5.
It is worth noting that there is a typo in the in-vitro parameters citation. Emax 10.7 and EC50 9.8 for CYP2B6 are mentioned in the literature, Hariparsad et al, 2008. The correct reference citation should be shown in the below screenshot. To double check these values, we digitized the literature data from Sahi et al. 2019, re-calculated the EC50 and Emax values using R, and summarized the results in the table below.
Original plot: CYP2C9 activity determined in microsomes prepared from following treatment with 0.4–400 μM phenytoin for 72 h.
Summary of results:
Calculated by the average value of the Enzyme activity fold
Indmax
IndC50
reference
1.9
15.3
Almond et al.,2016
Ban Ke A et al 2013
2.05
15.58
calculated by R
Line 139: “B:P” should be Rbp.
- Response: We thank reviewer for pointing this out. We updated the manuscript accordingly.
- Line 140: blood. What and how blood parameters were used in Kp prediction in Lukacova-method?
- Response: We thank the reviewer for highlighting this. The wording in the manuscript was confusing and we updated it accordingly to improve clarity. The method uses the blood parameters (specifically, the composition of red blood cells) in the Kp calculation in general, but that part of the calculation would not impact Kp prediction for acidic compounds like phenytoin.
Figure 2: Omeprazole is also metabolized by 3A4. Has the model considered the impact of phenytoin-induced 3A4 induction on the PK of omeprazole?
- Response: One of the limitations of the current analysis is that the software does not allow both victim and perpetrator compounds to act as inducers. However, in the clinical study evaluating the effect of the interaction of omeprazole on phenytoin (Prichard et al 1987) was done with phenytoin given only as single dose after seven days of omeprazole treatment. The induction effect of phenytoin would be apparent only after multiple doses (of phenytoin) and thus, would not be present in the study used for verification of the DDI with omeprazole. Therefore, even if the induction effect of phenytoin on 3A4 was not considered here, the study is still valid for the purpose of the DDI verification.
- Figure 6: For all DDI simulations, please provide semi-log plots as well, such that the elimination phase and the impact due to CYP-mediated DDI can be well demonstrated.
Response: We appreciated your suggestions, `and these were added (See Figure 6S of Supplementary Material).
- Figure 6(b): any comments on the model performance with respect to suspension formulation?
Response: The clinical study from Touchette et al. 1992 was used for verification purposes only. This study tested a single 250 mg dose of phenytoin suspension (125mg/5mL, Dilantin; Parke-Davis) in 9 healthy volunteers. The prediction of AUC ratio (AUC simulated/Observed) in this study was outside the 2-fold error. In addition, the observed exposure for this formulation was significantly higher than exposures reported for the 248 and 250 mg capsule ant tablet administrations suggesting that this might be due to formulation difference. Given that phenytoin is a BCS class II drug (i.e., low soluble and highly permeable drug), the absorption might be enhanced in the suspension formulation. This has been discussed in the main manuscript (see line 529). However, the model is able to predict the DDI ratio thus supporting the ability of the model of predicting the interaction.

Reviewer 2 Report
In the article "Comprehensive Physiologically-Based Pharmacokinetic Model to Assess Drug-Drug Interactions of Phenytoin" the authors presented well PBPK model development for Phenytoin with potential use in DDI assessment. Methods are well described, and results and discussion clearly presented.
The only point I have relates to plots 3 and 6. Could you reduce the width of the lines and make the points smaller? It that form the image is difficult to "read". My perception ("not correct") is like in business presentations someone is trying to "hide" some real differences/distances - no need for that in scientific work.
Just a comment for the future work - it could be great to have this model exported in R :)
Author Response
The only point I have related to plots 3 and 6. Could you reduce the width of the lines and make the points smaller? It that form the image is difficult to "read". My perception ("not correct") is like in business presentations someone is trying to "hide" some real differences/distances - no need for that in scientific work.
Response: We appreciated your suggestions, and these were fixed.
-Just a comment for the future work - it could be great to have this model exported in R :)
Response: Thanks to reviewer 2 for the comment. We’ll consider it in the next steps.

Reviewer 3 Report
In my opinion this manuscript can be published in present form.
Author Response
In my opinion this manuscript can be published in present form.
Response: Thank you, reviewer 3, for the positive review.

Reviewer 4 Report
This study described the physiologically based pharmacokinetic model to assess drug-drug interactions of phenytoin. The Gastroplus simulation software was used to develop model by assessing pharmacokinetic data of 15 different studies in healthy subject. Over all the manuscript is written well and within the scope of the journal. The data is presented systemically and supported by various graphs in models. Some minor correction is required before its consideration in pharmaceutics journal.
1. Abstract: so many abbreviations are mentioned in abstract. It’s better to avoid abbreviation in abstract section otherwise if mentioned, kindly used full word. e.g. FDA, EMA, CYP.
2. Pharmacokinetic interaction of phenytoin with other drugs (available literatures) should be mentioned in introduction section of the manuscript.
3. The source of Gastroplus simulation software (name, country) should be incorporate in the section 2.1.
4. Line 51-52 - It has to recheck the objective of the study. Is this model was developed as phenytoin is inducer of CYP3A4 or due to substrate of CYP2C9/CYP2C19?
5. It is suggested to re-organize the conclusion section much better. The conclusion is lacking some basic components. Kindly summarize as the problem(s), objectives methodology, findings, and recommendation(s).
Author Response
-
- Abstract: so many abbreviations are mentioned in abstract. It’s better to avoid abbreviation in abstract section otherwise if mentioned, kindly used full word. e.g. FDA, EMA, CYP.
- Response: Thanks for the suggestion, we have updated the abstract accordingly.
.
- Pharmacokinetic interaction of phenytoin with other drugs (available literatures) should be mentioned in introduction section of the manuscript.
Response: Thank you for your suggestion. We added it (See line 63).
-Patients undergoing chronic phenytoin treatment are at risk of DDIs when introducing medications primarily metabolized by CYP3A4 or medications that function as CYP2C9 inducers or inhibitors. Phenytoin can enhance the metabolism of co-administered CYP3A4 substrates, including estrogens[13], progestogens [13], voriconazole [14]], itraconazole [15], amiodarone [16], ritonavir [16], lopinavir [17], ivabradine [18], atorvastatin [19], nisoldipine [20], midazolam [21], quetiapine [22] digoxin [23], and cyclosporine [24]. Conversely, when combined with CYP2C9 and/or CYP2C19 inhibitors, such as, fluconazole [25,26], and voriconazole [14], phenytoin blood levels will increase, elevating the risk of side effects. Lopinavir [17] and ritonavir [17], through CYP2C9 induction, can have the opposite effect, reducing phenytoin blood levels and potentially increasing the risk of seizures.
The source of Gastroplus simulation software (name, country) should be incorporate in the section 2.1.
Response: Thanks. We added it. (See line 94)
- Line 51-52 - It has to recheck the objective of the study. Is this model was developed as phenytoin is inducer of CYP3A4 or due to substrate of CYP2C9/CYP2C19?
Response: We agree with the reviewer the objective was unclear. We have updated the objective (see line 72-74)
- It is suggested to re-organize the conclusion section much better. The conclusion is lacking some basic components. Kindly summarize as the problem(s), objectives methodology, findings, and recommendation(s).
Response: We appreciate the suggestions. We have updated the conclusions.

Reviewer 5 Report
The authors provided convincing results for their PBPK model for phenytoin and discussed transparently possible reasons for discrepancy between simulation and observed results. Highly relevant for this research field.
Author Response
The authors provided convincing results for their PBPK model for phenytoin and discussed transparently possible reasons for discrepancy between simulation and observed results. Highly relevant for this research field.
Response: Thank you, reviewer 5, for the positive review.

Round 2
Reviewer 1 Report
Thanks the authors for addressing my comments.